# The Local and Electronic Structure Study of Lu*_x_*Gd_1−*x*_VO_4_ (0 ≤ *x* ≤ 1) Solid Solution Nanocrystals

**DOI:** 10.3390/nano13020323

**Published:** 2023-01-12

**Authors:** Yang Chen, Ziqing Li, Nianjing Ji, Chenxi Wei, Xiulan Duan, Huaidong Jiang

**Affiliations:** 1State Key Laboratory of Crystal Materials, Institute of Crystal Materials, Shandong University, Jinan 250100, China; 2Institute of Optoelectronics, Fudan University, Shanghai 200433, China; 3School of Materials Science & Engineering, Shandong Jianzhu University, Jinan 250101, China; 4Center for Transformative Science, ShanghaiTech University, Shanghai 201210, China

**Keywords:** solid solution, local structure, electronic structure

## Abstract

Rare-earth-doped mixed crystals have demonstrated tunable optical properties, and it is of great importance to study the structural characteristics of the mixed-crystal hosts. Herein, Lu*_x_*Gd_1-*x*_VO_4_ (0 ≤ *x* ≤ 1) solid solution nanocrystals were synthesized by a modified sol–gel method, with a pure crystalline phase and element composition. The X-ray diffraction (XRD) and Rietveld refinement results showed that Lu*_x_*Gd_1−*x*_VO_4_ nanocrystals are continuous solid solutions with a tetragonal zircon phase (space group *I*4_1_/*amd*) and the lattice parameters strictly follow Vegard’s law. The detailed local structures were studied by extended X-ray absorption fine structure (EXAFS) spectra, which revealed that the average bond length of Gd-O fluctuates and decreases, while the average bond length of Lu-O gradually decreases with the increase in Lu content. Furthermore, the binding energy differences of core levels indicate that the covalent V-O bond is relatively stable, while the ionicity of the Lu-O bond decreases with the increasing *x* value, and the ionicity of the Gd-O bond fluctuates with small amplitude. The valence band structures were further confirmed by the first-principles calculations, indicating that the valence band is contributed to by the O 2*p* nonbonding state, localized Gd 4*f* and Lu 4*f* states, and the hybridized states between the bonding O 2*p* and V 3*d*. The binding energies of the Lu core and the valence levels tend to decrease gradually with the increase in Lu content. This work provides insight into the structural features of mixed-crystal hosts, which have been developed in recent years to improve laser performance by providing different positions for active ions to obtain inhomogeneous broadening spectra.

## 1. Introduction

Benefiting from excellent optical, electronic, magnetic, and thermal properties, rare-earth orthovanadates (REVO_4_, RE = Sc, Y, and La-Lu) have wide applications in many fields, such as phosphors, laser hosts, scintillators, catalysts, polarizers, photovoltaic cells, dielectric ceramics, magnetocaloric materials, photothermal therapy, and optical probes [1,2,3,4,5,6,7,8,9]. Due to the strong UV absorption of VO_4_^3−^ groups and the efficient energy transfer from VO_4_^3−^ groups to emitting ions, REVO_4_ materials have been used as highly efficient host materials [10]. Their similar ionic radii are conducive to the doping of some active ions, such as all lanthanide active ions, Bi^3+^, etc. Due to the abundant energy levels of 4*f*^n^ configurations, the 4*f*-4*f* and 5*d*-4*f* transitions of lanthanide active ions can emit light from ultraviolet to infrared [11]. In recent years, nanosized REVO_4_ materials with well-controlled shape, size, phase purity, and different active ions and doping concentrations have been widely studied to learn their luminescent properties and have been identified as promising materials for a wide range of optical applications [12,13,14,15,16]. In order to further expand the optical applications of REVO_4_-series materials, the influence of mixed-crystal hosts on the luminescence properties has also been explored. For example, Robinson et al. proved that Bi_1−*x*_Dy*_x_*VO_4_ showed broader XRD peaks compared to DyVO_4_, and a progressive decrease in the bandgap with the increase in Bi concentration and crystal field effects should not be negligible for the 4*f*-4*f* transitions [17]. Kang et al. confirmed that the adjustment of the cation fractions could tune the excitation tail and emission band maximum of Bi^3+^ ions through bandgap modulation in the (Y,Sc)VO_4_ host [18]. For Nd:Lu*_x_*Y_1−*x*_VO_4_ mixed crystals, although the absorption and emission peak locations were unchanged, the full width at half-maximum (FWHM) exhibited obvious inhomogeneous broadening compared to the two single crystals, meaning better Q-switched and mode-locked laser performance [19].

As mentioned above, the doped mixed-crystal materials may also have better performance than the single-crystal materials, besides the tunability of the physical properties of the materials. Thus, it is of great importance to study the structural characteristics of the mixed-crystal hosts. Among all of the REVO_4_ nanomaterials, GdVO_4_ and LuVO_4_ have been extensively studied for their unit cell structures, properties, and applications. They both possess a tetragonal zircon structure, belong to the space group *I*4_1_/*amd*, and can be used as catalysts and as hosts for luminescent materials [1,4,6,20,21,22,23,24]. Mixed crystals of the two orthovanadates Lu*_x_*Gd_1−*x*_VO_4_ doped with Nd^3+^ have been grown by the Czochralski method, and the XRD results revealed that GdVO_4_ and LuVO_4_ can form solid solutions in any proportion and maintain the zircon-phase structure [25]. By changing the composition of the solid solutions, the basic physical properties of the materials can be adjusted. Yu et al. demonstrated that the thermal, optical, and laser properties of Nd:Lu*_x_*Gd_1−*x*_VO_4_ show variation as a function of *x* [26]. Additionally, compared to pure crystals, the laser performance of mixed crystals is improved, due to the inhomogeneous broadening of the fluorescence lines generated by the mixed-crystal hosts, which indicate that although the spectroscopic and laser properties are mainly determined by the eigen multiplet of the active ions, they are also partly influenced by the crystal field [2,27]. As far as we know, there have been some previous reports on the growth and properties of large rare-earth-ion-doped Lu*_x_*Gd_1−*x*_VO_4_ crystals, but the local and electronic structures of nanosized Lu_x_Gd_1−x_VO_4_ solid solution materials are less studied.

It is known that the properties of a material are strongly dependent on its lattice structure. In the unit cell of zircon-type Lu*_x_*Gd_1−*x*_VO_4_, the Gd (Lu) atom coordinates with eight oxygen atoms to form an irregular dodecahedron with four longer and four shorter chemical bonds and occupies *D*_2*d*_ sites, while the V atom coordinates with four oxygen atoms to form a regular tetrahedron [28]. Active ions can substitute Lu^3+^ or Gd^3+^ to occupy the center of GdO_8_ or LuO_8_ dodecahedrons. Previous reports suggested that the reason for the inhomogeneous broadening of the spectral peaks is that active ions are randomly distributed in mixed-crystal hosts, occupying different lattice locations and experiencing different local crystal fields, so the spectral peaks are contributed by many different structural centers [19]. Therefore, compared with GdVO_4_ and LuVO_4_, the application of Lu*_x_*Gd_1−*x*_VO_4_ nanomaterials can be expanded to present as promising host materials.

In this work, a series of Lu*_x_*Gd_1−*x*_VO_4_ (0 ≤ *x* ≤ 1) nanocrystals were prepared by a simple sol–gel method. The changes to the unit cell, local and electronic structures, and composition were studied by experimental and theoretical methods. In terms of experimental technology, we combined X-ray powder diffraction (XRD), extended X-ray absorption fine structure (EXAFS), and X-ray photoelectron spectroscopy (XPS) to study the unit cell structures, the local structures of the Lu^3+^ and Gd^3+^ ions, the chemical states of elements, the covalency or ionicity of metal–oxygen (M–O) chemical bonds, and the valence band structures of the series solid solutions. The valence band structures were also investigated by comparison with the density of electronic states (DOS) calculated by the LSDA + *U* method. Our findings could provide a comprehensive understanding of the mixed-crystal structures of Lu*_x_*Gd_1−*x*_VO_4_ nanocrystals, which will be helpful to explore more applications.

## 2. Materials and Methods

A series of Lu*_x_*Gd_1−*x*_VO_4_ (*x* = 0, 0.1, 0.2, 0.3, 0.4, 0.5, 0.6, 0.7, 0.8, 0.9, 1) nanocrystals were synthesized by the sol–gel method. Stoichiometric Gd_2_O_3_ and Lu_2_O_3_ (Aladdin, 99.9%) were dissolved in diluted HNO_3_ solution under heating and stirring to obtain a transparent nitrate solution. Then, citric acid (C_6_H_8_O_7_·H_2_O, Sinopharm, AR) with a molar ratio of 2:1 to metal cations (Gd^3+^ and Lu^3+^ ions) was added as the chelating agent. After the citric acid was completely dissolved, ammonium metavanadate solution obtained by dissolving NH_4_VO_3_ (Sinopharm, AR) in hot deionized water was added dropwise to the above solution. After stirring and heating, viscous gels formed. The gels were then dehydrated at 100 °C for 6 h and calcined in air at 800 °C for 6 h to obtain pale-yellow powders with a heating and cooling rate of 50 °C/h. The powders were then finely ground for subsequent experiments.

The series of powder samples were identified by powder X-ray diffraction performed on a Bruker D8 advance powder diffractometer with Cu K_α1,2_ radiation (λ_average_ = 1.54184 Å). The molar contents of the constituent elements were measured using an EPMA-1720H (Shimadzu, Japan) Electron Probe Microanalyzer with the following parameters: AccV = 15.0 kV, BC = 9.9 nA, beam size = MIN, and SC = 6.8 nA. Additionally, YVO_4_, Gd_3_Ga_5_O_12_, and LuSiO_5_ were used as standard samples for the quantitative analysis of V, Gd, and Lu, respectively.

The local structures of the cations were investigated by extended X-ray absorption fine structure (EXAFS) measured at the beamline 14W1 of Shanghai Synchrotron Radiation Facility (SSRF, Shanghai, China). Gd L_3_-edge X-ray adsorption spectra at 7243 eV and Lu L_3_-edge X-ray adsorption spectra at 9244 eV of the samples were recorded in transmission and Lytle fluorescence (when the concentration was too low) modes at room temperature. The EXAFS spectra were normalized using Athena software, and the corresponding Fourier-transformed (FT) k^3^χ(k) plots were obtained by selecting the Hanning window. The FT plots were fitted using Artemis with k^3^-weighting to determine the distances between the central atoms and their coordination shells.

X-ray photoelectron spectroscopy (XPS) was performed using an ESCALAB 250XI spectrometer (Thermo Fisher Scientific) with monochromatized Al K_α_ X-ray radiation (1486.6 eV) in an ultrahigh vacuum (<10^−7^ Pa). A flood gun was used for charge neutralization. Survey scans were taken between 0 and 1300 eV with an energy step of 1 eV, fine scans of the elements’ characteristic peaks were taken with an energy step of 0.1 eV, and the valence bands were recorded between 0 and 37 eV with an energy step of 0.05 eV. All data were analyzed using the version 5.9918 of Thermo Avantage software.

Electronic structure calculations of Lu_0.5_Gd_0.5_VO_4_ (Appendix A) were performed using a LSDA + *U* approach [29] based on density functional theory (DFT) [30] implemented through the Cambridge Serial Total Energy Package (CASTEP) [31] program. The Hubbard-*U* corrections were introduced because simple local density approximation (LDA) [32] or generalized gradient approximation (GGA) [33] failed to describe the strong correlation between the highly localized 4*f* electrons of Gd and predicted a metallic band structure for GdVO_4_, which is inconsistent with its insulating nature. The calculations with several *U* values showed that Hubbard *U* has a significant impact on the Gd 4*f* energy level and the bandgap, and the Gd 4*f* is located at a deeper energy level when the *U* value is large. The *U* value (0.01 eV) added to the Gd 4*f* electrons was finally determined semi-empirically by comparing it to the XPS valence band spectra, but the calculated Gd 4*f* level was still deeper than the experimental binding energy. Vanderbilt-type ultra-soft pseudo-potentials were adopted to describe the electron–ion interactions. The atomic levels 4f^14^5p^6^5d^1^6s^2^ of the Lu atom, 4f^7^5s^2^5p^6^5d^1^6s^2^ of the Gd atom, 3s^2^3p^6^3d^3^4s^2^ of the V atom, and 2s^2^2p^4^ of the O atom were treated as valence electrons. The plane-wave cutoff energy of 600 eV was chosen for the calculations of the energy and properties. The (3 × 3 × 3) Monkhorst-Pack grid was used for Brillouin zone integrations.

## 3. Results and Discussion

### 3.1. Synthesis and Characterization of Lu_x_Gd_1−x_VO_4_ Nanocrystals

#### 3.1.1. XRD and Rietveld Refinement

All Lu*_x_*Gd_1−*x*_VO_4_ powder samples had a pure tetragonal zircon-phase structure with no impurity phase as identified by X-ray diffraction (Figure 1), which is indicative of continuous solid solutions. In the inset in Figure 1, a gradual shift to high angles for the positions of the diffraction peaks with increasing Lu content can be seen clearly, which is consistent with the trends of bulk mixed crystals [25]. This shift is probably due to the substitution of Lu^3+^ with a smaller ionic radius for Gd^3+^. The diffraction peaks of the Lu*_x_*Gd_1−*x*_VO_4_ (0 < *x* < 1) solid solutions show a broadening compared to the peaks of the two end members, which is also suggestive of compositional inhomogeneity due to the mismatch in radius between Lu^3+^ and Gd^3+^ ions.

Based on the XRD data, the unit cell parameters *a* and *c*, along with the unit cell volume (*V*) of the Lu*_x_*Gd_1−*x*_VO_4_ samples, were calculated by Rietveld refinement using FullProf Suite software (Appendix A). They all decreased linearly with the increase in Lu content, as can be seen from Figure 2, which conforms to Vegard’s law [34]. When *x* changes from 0 to 1, the values of *a*, *c*, and *V* decrease by 2.6%, 1.9%, and 6.8%, respectively, showing anisotropy.

#### 3.1.2. EPMA

EPMA was used to determine the actual compositions of the series of Lu*_x_*Gd_1−*x*_VO_4_ (*x* = 0.1, 0.3, 0.5, 0.7, 0.9) samples. It can be seen from Table 1 that the molar ratios of Lu/RE were very close to the nominal ratios (*x* values); therefore, *x* values are used to denote the actual Lu contents in this work. Because only metal elements were measured, V molar contents close to 50% for all of the samples indicate that they were also consistent with the stoichiometric ratios.

#### 3.1.3. TEM

The morphology and grain sizes of the Lu*_x_*Gd_1−*x*_VO_4_ nanocrystals were observed from the TEM images (Figure 3). The crystalline grains of all of the samples were irregular in shape and easily agglomerated. Under the same calcination temperature and time, the grain size of these nanocrystals decreased with increasing *x* values, and the average grain sizes were about 30–50 nm. Therefore, continuous Lu*_x_*Gd_1−*x*_VO_4_ (0 ≤ *x* ≤ 1) solid solution nanocrystals with a tetragonal zircon-type structure were successfully synthesized by the sol–gel method.

### 3.2. Local Structure of Lanthanide Atoms in Lu_x_Gd_1−x_VO_4_ Solid Solutions

In order to further study the linear evolution of the unit cell parameters of the Lu*_x_*Gd_1−*x*_VO_4_ solid solutions caused by cation substitution, the local structures of the lanthanide cations were investigated by EXAFS. Because they share the same zircon-type structure, the atomic distributions of the coordination shells of Gd atoms in GdVO_4_ and Lu atoms in LuVO_4_ are similar. The first, second, and third coordination shells of Gd/Lu atoms consist of eight O atoms, two V atoms, and four Gd/Lu and four V atoms, respectively, as shown in Figure 4. More precisely, the first coordination shell consists of two subshells, with the same coordination number (4) and a bond length difference of 0.12 Å for GdVO_4_ and 0.15 Å for LuVO_4_. The interatomic distances of the first two coordination shells from the central atoms were obtained by fitting the Fourier-transformed (FT) EXAFS spectra; the values, along with the fitting parameters and the average bond lengths of the two subshells, are listed in Appendix A.

As shown in Figure 5, except for the samples with the lowest elemental contents (*x* = 0.9 in Figure 5a and *x* = 0.1 in Figure 5b), the FT-EXAFS spectra of the series of solid solutions showed little change in the range of the first two coordination shells. The change in the peak shape of the first coordination shell was probably due to the change in the distortion index of the REO_8_ dodecahedron (Appendix A). The average Gd-O bond lengths reduced from 2.398 to 2.357 Å as *x* increased from 0 to 0.9, while the average Lu-O bond lengths reduced gradually from 2.356 to 2.314 Å as *x* increased from 0.1 to 1 (Figure 6). This suggests that the irregular REO_8_ dodecahedron is not rigid enough to remain unchanged in Lu*_x_*Gd_1−*x*_VO_4_ solid solutions but will shrink or relax slightly with the change of the unit cell. Although the Gd-O and Lu-O bond lengths did not follow Vegard’s law, the average RE-O bond lengths were distributed near the linear fits, following Vegard’s law. The Gd-V interatomic distances reduced gradually while the Lu-V interatomic distances first increased and then decreased as the Lu content increased. Moreover, the average RE-V interatomic distances followed Vegard’s law.

### 3.3. X-ray Photoelectron Spectroscopy of Lu_x_Gd_1−x_VO_4_ Solid Solutions

The elements, chemical states, and valence band structures of the Lu*_x_*Gd_1−*x*_VO_4_ nanocrystals were studied by XPS. The survey scans (Figure 7) confirmed the existence of only Lu, Gd, V, and O in all samples, except for the ubiquitous contaminated carbon. The intensity of characteristic peaks of Gd 3*d*, 4*d* and Lu 4*p*, 4*d* was enhanced with the increase in their respective contents; that is, they varied inversely with the change of *x*.

#### 3.3.1. Chemical States of O, V, Lu, and Gd in Lu_x_Gd_1−x_VO_4_ Solid Solutions

The core-level O 1*s*, V 2*p*, Gd 4*d*, and Lu 4*d* XPS spectra (Figure 8, Figure 9 and Figure 10) were analyzed to determine the chemical states of these elements in the Lu*_x_*Gd_1−*x*_VO_4_ solid solutions. The C 1*s* spectra all showed two peaks with a separation of about 4.6 eV, in which the stronger C 1*s* peak (284.6 eV) of the adventitious carbon (C-C/C-H) was used as a reference for binding energy calibration [35].

Due to the small overlap between the O 1*s* and V 2*p*_1/2_ peaks, it was essential to analyze the O 1*s* and V 2*p* spectra together [36]. The V 2*p* spectra contained two peaks due to spin–orbit splitting—an asymmetrical 2*p*_1/2_ and a symmetrical 2*p*_3/2_—and the 2*p*_1/2_ peak was broadened because of the Coster–Kronig effect [37]. The O 1*s* peaks were located at the higher-energy side of V 2*p*_1/2_ peaks. All of these O 1*s* and V 2*p* spectra showed a similar shape but a small amount of peak position shift, as can be seen from Figure 8a. The binding energy of O 1*s* changed from 529.9 to 530.2 eV, and it first increased and then decreased with the increases in *x* (Appendix A). In addition to the predominant lattice O 1*s* peak, a low-intensity component located at 1.5 eV higher (inset of Figure 8a) could be attributed to the surface-adsorbed hydroxyl groups [38,39,40]. The binding energy of V 2*p*_3/2_ changed from 517.1 to 517.4 eV, similar to the change in the lattice O 1*s* (Figure 8b and Appendix A). All of the binding energies were about 517.4 eV, indicating that V ions have only one +5 valence state in these solid solutions [36,41,42].

All of the Lu 4*d* spectra (Figure 9a) exhibited a doublet character due to the spin–orbit interaction. The 4*d*_3/2_ and 4*d*_5/2_ peaks had almost the same FWHM, due to the 4*f*^14^ filled shell of Lu^3+^, whereas metal Lu 4*d* peaks usually show extra broadening [43]. A separation of about 9.9 eV was observed, and the peak area ratios were in the range of 0.60–0.69—close to the theoretical values of 10.00 eV and 2/3, respectively [44]. The binding energy of the Lu 4*d*_5/2_ peak decreased gradually from 196.8 to 196.1 eV as *x* increased from 0.2 to 1 (Figure 9b and Appendix A).

Meanwhile, in the case of the Gd 4*d* core level, due to the existence of strong Coulomb-exchange interactions between 4*d* and the half-filled 4*f*^7^ states together with the spin–orbit interaction in the 4*d* state, the Gd 4*d* XPS spectra exhibited a broad multiplet splitting structure including four peaks (A, B, C and D), as shown in Figure 10a [45]. Kowalczyk et al. qualitatively described the multiplet with spin antiparallel ^7^D*_J_* (*J* = 1, …, 5) (peaks B, C and D) and spin parallel ^9^D*_J_* (*J* = 2, …, 6) (peak A) final states [46]. In this work, the ^9^D states were fitted by five sharp peaks with the same FWHM (1.66–1.83 eV) and a separation of 1.1 eV, and peak B was fitted by a broad Lorentzian–Gaussian mixed peak, according to the high-resolution XPS and lifetime broadening effect [41,45,47]. The fitting peaks of partial Gd 4*d* of GdVO_4_ are shown in the inset of Figure 10a as an example. The ^9^D_5_ states (141.5–141.8 eV) with the strongest intensity were selected to observe the evolution of the binding energies of the Gd 4*d* level. The binding energies of Gd 4*d* showed a 0.1–0.3 eV chemical shift with the variation in the Lu content (Figure 10b and Appendix A).

As mentioned above, changes existed in the core-level binding energies of the elements in the series of solid solutions. Furthermore, the binding energy difference ΔBE(O-M) = BE(O 1*s*) − BE(M core level) was used to characterize the ionicity or covalency of M-O chemical bonds [48]. Additionally, it was concluded that lower ΔBE(O-M) values are associated with stronger ionicity or weaker covalency of M-O bonds, while higher ΔBE(O-M) values indicate weaker ionicity or stronger covalency [49,50]. For the Lu*_x_*Gd_1−*x*_VO_4_ solid solutions in this work, the ΔBE values of O-V, O-Lu, and O-Gd are listed in Appendix A and shown in Figure 8b, Figure 9b, and Figure 10b, respectively.

The ΔBE(O-V) value between O 1*s* and V 2*p*_3/2_ for all of the samples was 12.8 eV, which is consistent with the reported values of vanadium pentoxide [36,51]. The unchanged ΔBE(O-V) values indicate that the ionicity and covalency of V-O chemical bonds are stable and that the VO_4_ tetrahedrons show strong rigidity in the series of solid solutions. This is probably due to the more covalent character of V-O bonds in GdVO_4_ and LuVO_4_ revealed by the analysis of electron density difference and Mulliken population analysis [52,53]. The ΔBE(O-Lu) values between O 1*s* and Lu 4*d*_5/2_ tend to increase as the Lu content increases, indicating the decrease in the ionicity of the Lu-O chemical bonds with the increasing *x*. Meanwhile, the ΔBE(O-Gd) values between O 1*s* and Gd 4*d* show volatile changes with small amplitude, indicating that the ionicity of the Gd-O chemical bonds changes very little as the *x* increases.

#### 3.3.2. Valence Band Structure Analysis of Lu_x_Gd_1−x_VO_4_ Solid Solutions

The XPS valence band spectra (Figure 11a) of the samples were analyzed in comparison with the first-principles-calculated density of states (DOS) and partial density of states (PDOS) of Lu_0.5_Gd_0.5_VO_4_ (Figure 11b).

The valence band spectra (0–37 eV) are mainly composed of peaks of the Lu 5*p*, Lu 4*f*, Gd 5*p*, Gd 4*f*, O 2*s*, O 2*p*, and V 3*d* states, in which the doublet peaks at ~33.7 eV and ~27.3 eV are assigned to the Lu 5*p*_1/2_ and Lu 5*p*_3/2_ states [54]. The wide band ranging from 12.0 eV to 30.0 eV, with lower intensity, is attributed to the Gd 5*p* ^7^P final state and the hybridization between the Gd 5*p* ^9^P final state and the O 2*s* state [55]. The predominant peak with a bump on its lower-binding-energy side, ranging from 2.5 eV to 12.5 eV, is the valence band (VB). The strongest splitting peaks located at ~9.0 eV and ~7.8 eV, in the middle of the VB, should be assigned to Lu 4*f*_5/2_ and 4*f*_7/2_ doublets [17]. Meanwhile, the Gd 4*f* state presents as an asymmetric peak with much lower intensity located at ~8.2 eV, which is overlapped with the Lu 4*f* doublet and a bump (broad and very low intensity) of the hybridized states between the bonding O 2*p* and V 3*d*. Nevertheless, the Gd 4*f* and Lu 4*f* states are highly localized and do not participate in chemical bonding with ligand O atoms, as can be seen from the DOS and PDOS. The valence band maximum (VBM) is a bump and is contributed to by the O 2*p* nonbonding state. The hybridized states between the bonding O 2*p* and V 3*d* determine the minimum and width of the VB, leading to a broader and broader VB as the *x* value decreases. Due to the higher intensity and specific doublet shape, it is obvious that the binding energies of the Lu 5*p* and 4*f* levels tend to decrease gradually with the increase in the Lu content, which is consistent with the evolutionary trend of the binding energy of the Lu 4*d* core level. This gradual change in the binding energies of the Lu core and valence levels originates from the variations in the lattice parameters induced by the displacement of Lu^3+^ with Gd^3+^.

## 4. Conclusions

A series of Lu*_x_*Gd_1−*x*_VO_4_ (0 ≤ *x* ≤ 1) nanocrystals of 30–50 nm in size were synthesized by a citric acid sol–gel process. XRD demonstrated that the as-synthesized nanocrystals were single-phase zircon-type continuous solid solutions, and the lattice parameters decreased linearly with the increase in the *x* value—*a* decreased from 7.21287 Å to 7.02852 Å, while *c* decreased from 6.35552 Å to 6.23600 Å, following Vegard’s law. The local structures of the central atoms of the GdO_8_ and LuO_8_ dodecahedrons studied by EXAFS showed that, to adapt to the shrunk unit cell, the Gd-O interatomic distances fluctuate reduced and Lu-O interatomic distances gradually reduced as the *x* value increased. The core-level electronic structures of all of the composition elements in Lu*_x_*Gd_1−*x*_VO_4_, as functions of the *x* values, were analyzed by XPS. With the Lu content increasing from 0 to 1, the binding energy values of O 1*s* and V 2*p* first increased and then decreased, and their differences remained unchanged, indicating that the covalency of V-O bonds is less affected by composition. The ionicity of the Lu-O bonds weakened with the increasing *x*, while the ionicity of the Gd-O bonds fluctuated with small amplitude. The valence band electronic structure of Lu*_x_*Gd_1−*x*_VO_4_ was also studied by the combination of XPS and first-principles calculations. The O 2*p* nonbonding state was located at the top of the VB, the Gd 4*f* and Lu 4*f* states were highly localized and located in the middle of the VB, while the hybridized state between the bonding O 2*p* and V 3*d* was located at the bottom of the VB and overlapped with Gd 4*f* and Lu 4*f* states. The Lu 5*p*, 4*f* levels and 4*d* core level tended to decrease gradually with the increase in the *x* value. In conclusion, due to the lanthanide contraction, the Gd-O and Lu-O bond lengths decreased to adapt to the shrunk unit cell, while the V-O bond length remained unchanged, as the *x* value increased from 0 to 1 throughout the Lu*_x_*Gd_1−*x*_VO_4_ series. Additionally, the binding energy shift of Lu and the ΔBE(O-Lu)-evaluated ionicity of the Lu-O bonds presented a positive correlation with the Lu-O bond length. This work provides insight into the local and chemical environments of the central atoms of GdO_8_ and LuO_8_ dodecahedrons in Lu*_x_*Gd_1−*x*_VO_4_, that is, the different positions provided for the substituted active ions, which are conductive to understanding the mechanism of inhomogeneous spectral broadening of mixed-crystal hosts in laser applications.

## Figures and Tables

**Figure 1 nanomaterials-13-00323-f001:**
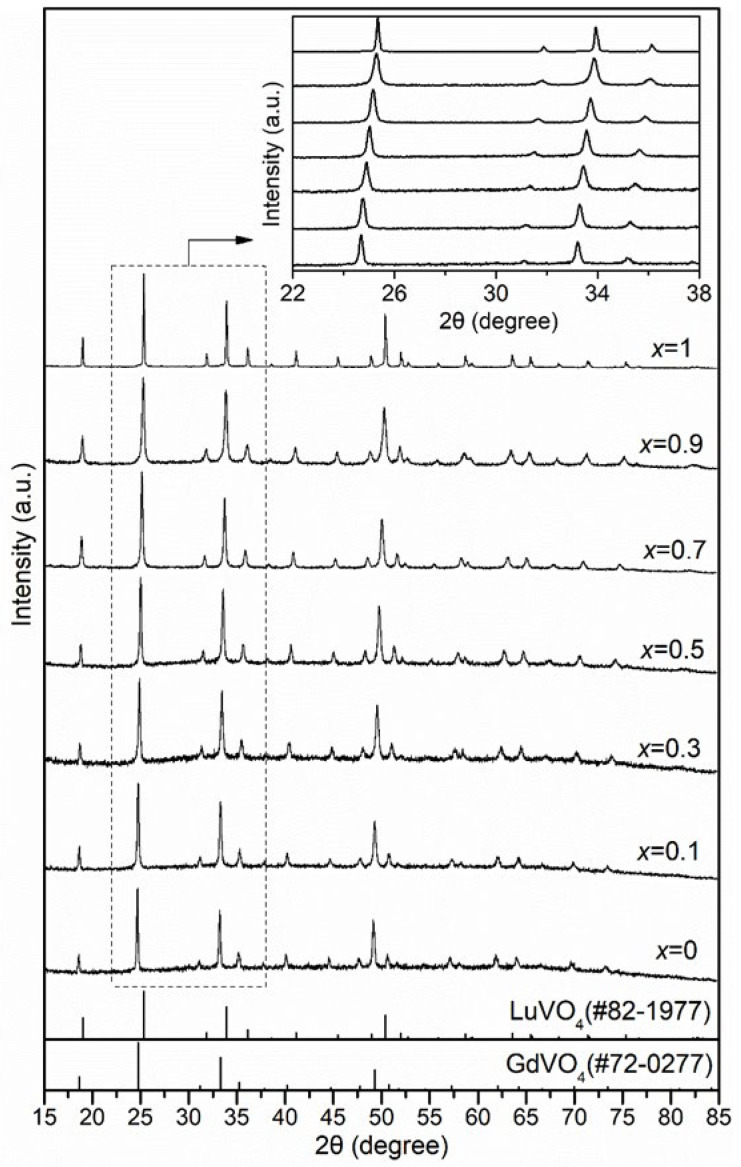
X-ray diffraction patterns of tetragonal Lu*_x_*Gd_1−*x*_VO_4_ with standard patterns of GdVO_4_ and LuVO_4_ shown at the bottom (vertical bar). The inset is an enlargement of the 2θ region between 22° and 38°.

**Figure 2 nanomaterials-13-00323-f002:**
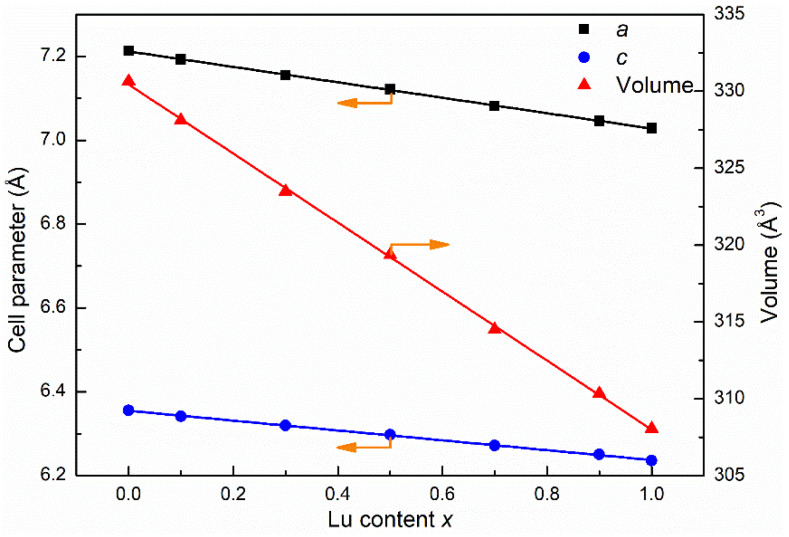
Lattice parameters *a*, *c*, and unit cell volume as functions of *x* values in Lu*_x_*Gd_1−*x*_VO_4_.

**Figure 3 nanomaterials-13-00323-f003:**
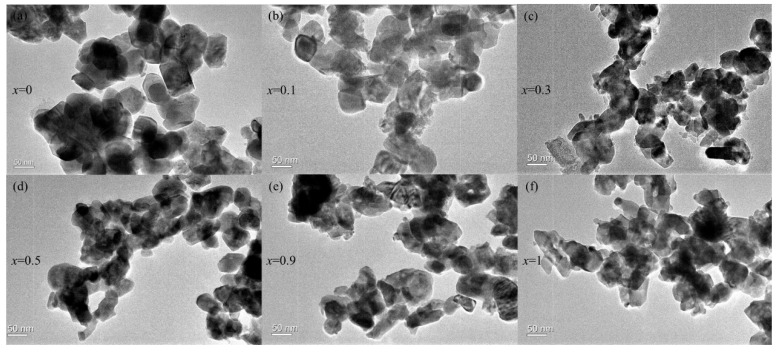
Transmission electron microscopy images obtained for (**a**) *x* = 0, (**b**) *x* = 0.1, (**c**) *x* = 0.3, (**d**) *x* = 0.5, (**e**) *x* = 0.9 and (**f**) *x* = 1 of Lu*_x_*Gd_1−*x*_VO_4_ nanocrystals.

**Figure 4 nanomaterials-13-00323-f004:**
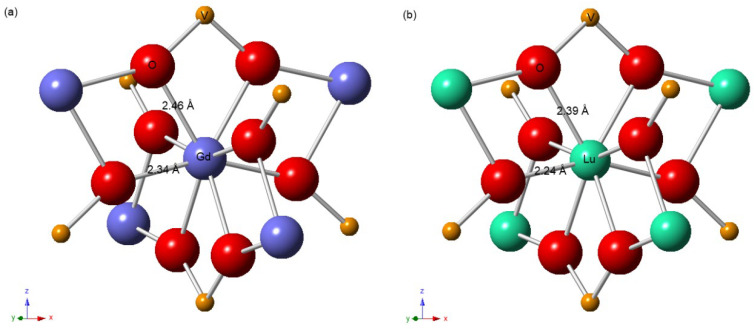
Distribution of neighboring atoms of the first three shells of the central (**a**) Gd atom in GdVO_4_ and (**b**) Lu atom in LuVO_4_ with the zircon-type structure; the annotated interatomic distances of the first coordination shell were obtained from EXAFS.

**Figure 5 nanomaterials-13-00323-f005:**
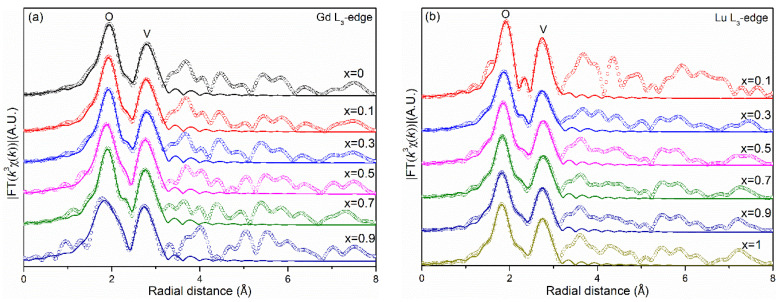
Magnitude of the Fourier-transformed EXAFS spectra (hollow circles) and the corresponding fitting curves (solid lines) of the first two coordination shells at the (**a**) Gd L_3_-edge and (**b**) Lu L_3_-edge for Lu*_x_*Gd_1−*x*_VO_4_ solid solutions.

**Figure 6 nanomaterials-13-00323-f006:**
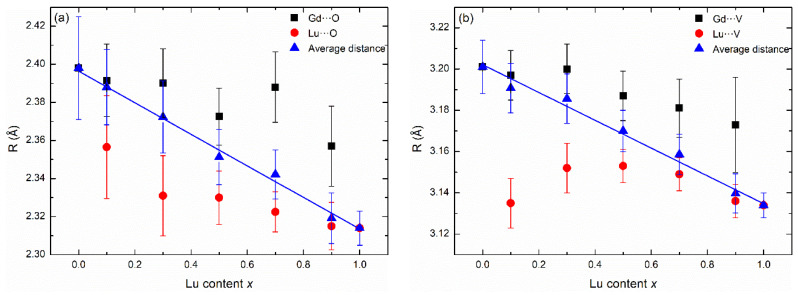
The (**a**) first and (**b**) second coordination shells’ interatomic distances with respective error bars of Gd^3+^ (square) and Lu^3+^ (dot), and the average distances (triangles) in the Lu*_x_*Gd_1−*x*_VO_4_ solid solutions. The average distances are weighted values considering the elemental contents, and the solid blue lines are the linear fits.

**Figure 7 nanomaterials-13-00323-f007:**
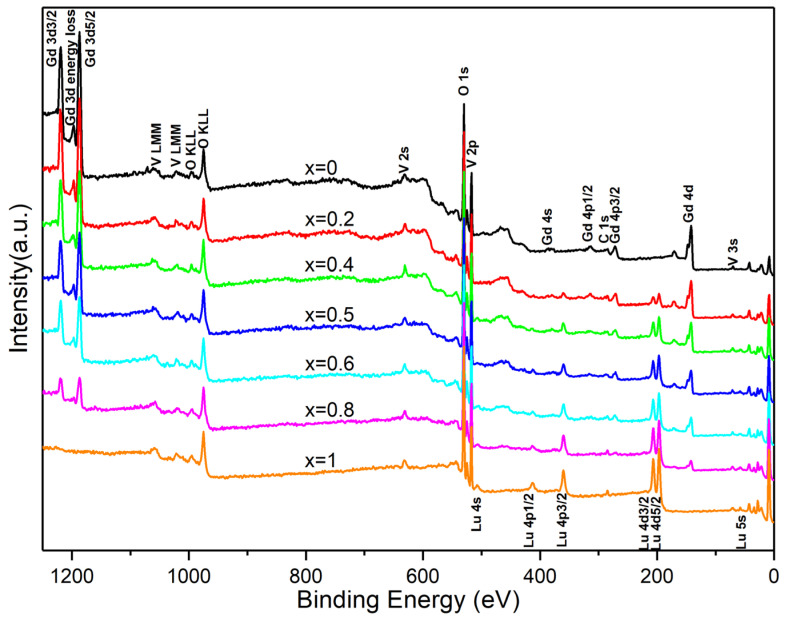
XPS survey spectra of Lu*_x_*Gd_1−*x*_VO_4_ (0 ≤ *x* ≤ 1) nanocrystals.

**Figure 8 nanomaterials-13-00323-f008:**
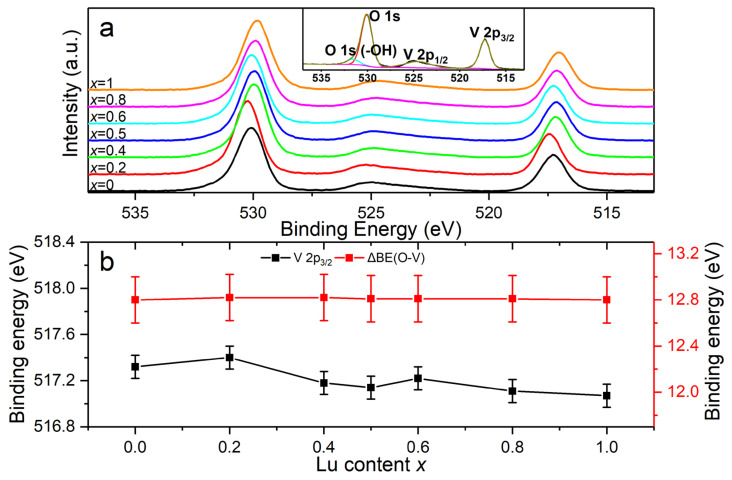
(**a**) Stacked O 1*s* and V 2*p* XPS spectra of Lu*_x_*Gd_1−*x*_VO_4_ (0 ≤ *x* ≤ 1) and the deconvoluted spectrum of GdVO_4_ (inset). (**b**) Binding energy of V 2*p*_3/2_ and binding energy difference between O 1*s* and V 2*p*_3/2_ versus the Lu content *x*. The error bars are 0.1 and 0.2 eV, respectively.

**Figure 9 nanomaterials-13-00323-f009:**
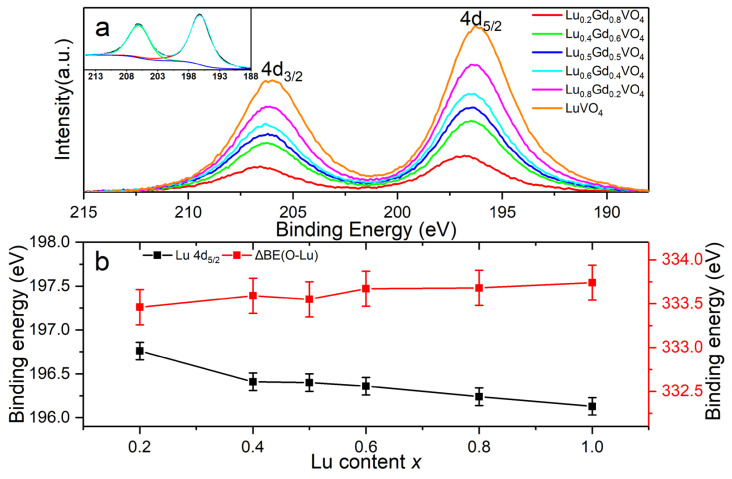
(**a**) Background-subtracted Lu 4*d* XPS spectra of Lu*_x_*Gd_1−*x*_VO_4_ (0 < *x* ≤ 1) and the doublet fitting Lu 4*d* spectrum of LuVO_4_ (inset). (**b**) Binding energy of Lu 4*d*_5/2_ and the binding energy difference between O 1*s* and Lu 4*d*_5/2_ versus the Lu content *x*. The error bars are 0.1 and 0.2 eV, respectively.

**Figure 10 nanomaterials-13-00323-f010:**
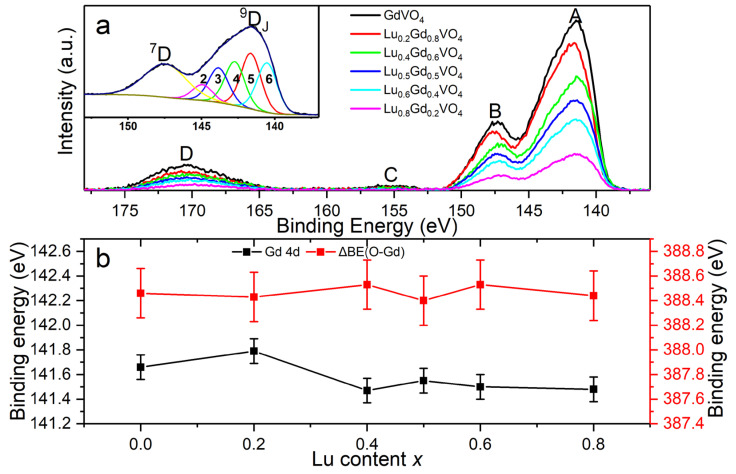
(**a**) Background-subtracted Gd 4*d* XPS spectra of Lu*_x_*Gd_1−*x*_VO_4_ (0 ≤ *x* < 1) and the deconvolution of peaks A and B of GdVO_4_ (inset). (**b**) Binding energy of Gd 4*d* ^9^D_5_ peaks and binding energy difference between O 1*s* and Gd 4*d* ^9^D_5_ versus the Lu content *x*. The error bars are 0.1 and 0.2 eV, respectively.

**Figure 11 nanomaterials-13-00323-f011:**
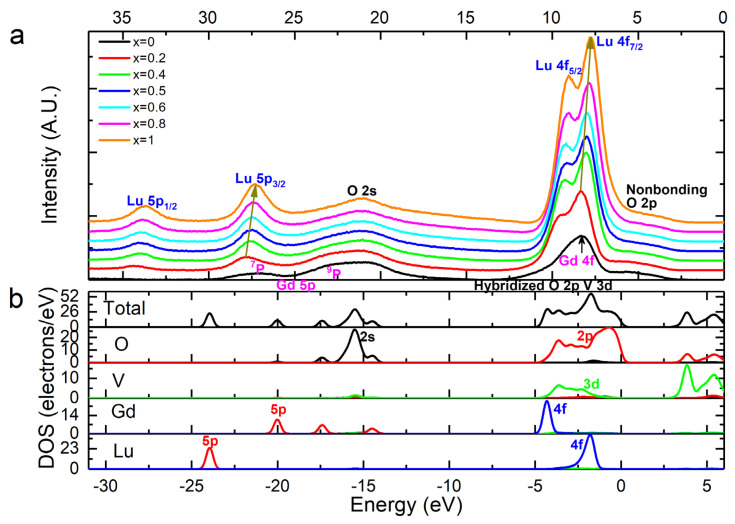
(**a**) Background-subtracted XPS valence band spectra of Lu*_x_*Gd_1−*x*_VO_4_ (0 ≤ *x* ≤1) solid solutions; (**b**) LSDA + *U*-calculated DOS and PDOS of Lu_0.5_Gd_0.5_VO_4_.

**Table 1 nanomaterials-13-00323-t001:** EPMA compositions of Lu*_x_*Gd_1−*x*_VO_4_ solid solutions.

x	0.1	0.3	0.5	0.7	0.9
V (mol %)	50.295	50.719	49.543	51.071	50.627
Gd (mol %)	44.910	34.566	25.230	15.590	4.960
Lu (mol %)	4.795	14.714	25.227	33.338	44.413
Molar ratio of Lu/RE ^1^	0.10	0.30	0.50	0.68	0.90

^1^ RE is the sum of Lu and Gd.

## Data Availability

Not applicable.

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
