# Peer review of "The Local and Electronic Structure Study of LuxGd1−xVO4 (0 ≤ x ≤ 1) Solid Solution Nanocrystals"

_nanomaterials, 2023, doi:10.3390/nano13020323_

Round 1
Reviewer 1 Report
I think the presented paper is excellent, both in merit and presentation. I have only minor remarks about the clarity of a few sentences:
1. Line 39: “As active ions, lanthanide ions have abundant emission colors in 39 the whole visible light range originating from their 4f-4f or 5d-4f transitions [11].” – this sentence is not clear, especially the statement of abundant emission colors
2. Line 73: “… but nanosized LuxGd1-xVO4 solid solution materials are few studied.” – the end of the sentence needs changing
3. Paragraph 94 - 95: “… the density of electronic states (DOS) calculated by LSDA+U method calculated.” – the calculated at the end of the sentence has to be omitted
Reviewer 2 Report
The article is devoted to the study of complexly substituted nanoparticles LuxGd1-xVO4 (0≤x≤1). It is shown that an unrestricted solid solution is formed in the LuVO4 - GdVO4 system. Structural features and band structure of the obtained mixed nanocrystals are studied.
The work may be published in the journal after minor corrections.
Abstract: it is necessary to add a space group for these crystals and a range of lattice parameters.
What was the motivation for this study? Why it was carried out and how the data obtained can be used for the practical application of these objects. It is necessary to add information at the end.
Line 55 - As mentioned above, the doped mixed crystal materials may also have better performance than the single crystal materials, except for the tunability of physical properties of 56 materials. T...
What is meant. The meaning of the sentence is not clear. Why "except for"? Solid solutions allow you to vary the properties over a wide range!
